# The Effects of Nature Exposure Therapies on Stress, Depression, and Anxiety Levels: A Systematic Review

**Diana Marcela Paredes-Céspedes [1],\*, Norida Vélez [1], Alejandra Parada-López [1], Yesith Guillermo Toloza-Pérez [1], Eliana M. Téllez [1], Claudia Portilla [1], Carolina González [2], Leany Blandón [2], Juan Carlos Santacruz [2] and Jeadran Malagón-Rojas [1]**

[1] Grupo de Salud Ambiental y Laboral, Instituto Nacional de Salud, Dirección de Investigación, Bogotá 111321, Colombia; noridavelezc@gmail.com (N.V.); mparada@ins.gov.co (A.P.-L.); yesith04@gmail.com (Y.G.T.-P.); etellez@ins.gov.co (E.M.T.); cportilla@ins.gov.co (C.P.); jmalagon@ins.gov.co (J.M.-R.)

[2] Fundación Colombiana del Corazón, Bogotá 111311, Colombia; cgonzalezd@ecotuconsulting.com (C.G.); lblandon@corazonesresponsables.org (L.B.); jcsantacruz@corazonesresponsables.org (J.C.S.)

\* Correspondence: dparedes@ins.gov.co; Tel.: +57-(601)-220-7700 (ext. 1624)

**Abstract:** Background: Mental well-being plays a pivotal role within the broader spectrum of health and illness, encompassing factors such as stress, depression, and anxiety. Nature-based therapeutic interventions have emerged as a promising approach to addressing these mental health challenges. This study seeks to assess the impact of these interventions on stress, depression, and anxiety levels. Methods: We conducted an extensive search for randomized clinical trials that examined stress, anxiety, and depression levels. The selected studies underwent a rigorous risk-of-bias assessment following the guidelines outlined in the Cochrane Handbook for Systematic Reviews. Results: Our review encompassed findings from eight publications. Among them, two studies measuring cortisol levels revealed significant differences between the pre-test and post-test measurements within the intervention groups. In two studies that employed the Stress Response Inventory, a significant decrease in stress levels was observed within the intervention groups in contrast to the control groups. However, no significant differences were noted in studies that utilized the Restorative Outcome Scale. In the assessment of anxiety and depression levels, three studies employed the Positive and Negative Affect Schedule, while four studies utilized The Profile of Mood States scale; none of these studies demonstrated significant differences. Conclusions: The current body of evidence offers limited support for advocating nature-based therapeutic interventions as a primary approach to reducing stress, depression, and anxiety.

**Keywords:** nature; stress; forests; therapy cortisol; blood pressure; intervention

## 1. Introduction

Psychological factors play a significant role in the development and progression of various health conditions [1]. Among these factors, emotional and psychological disorders such as stress, depression, and anxiety [2] are particularly noteworthy, posing global mental health challenges. These conditions have far-reaching implications, impacting chronic diseases like diabetes, cardiovascular conditions, cancer, and obesity, potentially affecting disease outcomes [3]. The alarming rise in stress, depression, and anxiety rates has become a pressing public health crisis, leading to increased suicide rates in many countries. According to the World Health Organization (WHO) [4], depression affects over 264 million people worldwide, making it one of the most prevalent mental disorders. Anxiety disorders are also common, affecting approximately 3.6% of the global population. Mental health conditions are a leading cause of disability globally, with depression and anxiety alone costing the global economy over USD 1 trillion in lost productivity each year. Beyond their economic toll, mental health issues have profound social implications, including disrupted

relationships, reduced educational and employment opportunities, and a higher likelihood of substance abuse [5].

Stress, the body's response to threats or pressure, can increase susceptibility to inflammatory disorders, including infectious diseases. Prolonged stress responses can lead to physiological changes, particularly in the brain, contributing to disease development [6]. Experts suggest that an individual's response to stress is intricately linked to their interactions within their social environment [7].

Anxiety, characterized by excessive fear and tension about potential threats, can vary from adaptive to pathological states. Persistent and debilitating anxiety levels associated with distress or impaired psychological functioning fall into the latter category [8], [9]. Globally, approximately 301 million individuals, equivalent to 4.05% of the world's population, grapple with anxiety disorders, marking a significant increase of over 55% from 1990 to 2019 [10].

In recent years, nature immersion therapies have emerged as alternative approaches to alleviating stress, depression, and anxiety [11–13]. These therapies leverage exposure to natural environmental stimuli to induce physiological relaxation, potentially enhancing immune functions and aiding in disease prevention [14,15]. Various methodologies for nature immersion therapy have been developed, encompassing practices such as Shinrin-Yoku, mindfulness, yoga, physical activity, and Tai-Chi in natural settings [16–18]. "Shinrin-Yoku", which translates to "absorbing the forest atmosphere through all senses", is commonly known as forest bathing. This therapeutic modality is associated with a myriad of positive health benefits for both physiological and psychological well-being [19,20].

Proponents of Shinrin-Yoku assert that its primary effects include enhancements to the immune system (including increased natural killer—NK cells and reduced allergies) and improvements in the cardiovascular system (notably reduced blood pressure) [21–23]. Nature immersion therapies are described to offer a range of psychological benefits, including stress reduction, anxiety alleviation, and depression mitigation. They provide a natural haven for mental relaxation, improved mood control, and enhanced immune function, offering potential benefits for conditions like attention deficit hyperactivity disorder and various other positive effects [21–23].

Regarding the evaluation of the human response to stress, there a myriad of instruments focused on assessing stress, anxiety, fatigue, and health-related effects in research and clinical practice. Some of the most widely recognized scales include the Profile of Mood States (POMS) [24] questionnaire that provides a multidimensional assessment of mood states, addressing transient emotional states such as tension/anxiety, depression/rejection, anger/hostility, vigor/activity, fatigue/inertia, and confusion/bewilderment. Additionally, a Total Mood Disturbance (TMD) score can be calculated by summing the scores from the first five subscales and subtracting the score of the vigor/activity subscale. In addition, the Positive and Negative Affect Schedule (PANAS) [11,25] evaluates positive and negative effects, providing insights into an individual's emotional balance. The Restorative Outcome Scale (ROS) [26] measures an environment's capacity to promote psychological restoration and stress relief. The Stress Response Inventory (SRI-MF) [27] focuses on assessing stress responses across various dimensions. Lastly, the 10-item Perceived Stress Scale (PSS10) [28] is used to gauge the perception of stress in everyday situations. These scales represent valuable tools for measuring and quantifying emotional well-being in various research and clinical settings.

Furthermore, there are alternative approaches to assessing the physiological response to stress, such as measuring the cortisol levels in the blood and the cortisol awakening response (CAR) [29]. However, these methods are less commonly employed due to their high costs and logistical demands.

Despite evidence suggesting the potential of nature immersion therapies in reducing stress, anxiety, and depression, conflicting results and significant methodological variations across studies persist. Therefore, this systematic review aims to comprehensively assess the impact of nature-based therapeutic interventions on stress, depression, and anxiety levels.

The primary driving force behind this study was the growing interest in nature-based therapeutic interventions and their potential impact on mental well-being. With increasing urbanization and the associated rise in stress, depression, and anxiety levels, understanding the effectiveness of nature therapies has become increasingly important. Our goal was to contribute to the existing body of knowledge by conducting a systematic review and meta-analysis to assess the available evidence on the effects of nature therapies on stress, depression, and anxiety. By synthesizing the findings from existing studies, we aimed to provide valuable insights for both researchers and practitioners in the field of mental health and well-being.

## 2. Materials and Methods

We carried out a systematic review in accordance with the guidelines of the Cochrane Collaboration and followed the Preferred Reporting Items for Systematic Reviews and Meta-Analyses (PRISMA) recommendations (PRISMA) [30]. Our review protocol was registered on the international prospective register for systematic reviews, PROSPERO (CRD42021279423).

### 2.1. Selection Criteria

Randomized clinical trials involving healthy participants aged 18 years or older were incorporated into this review. We encompassed studies that conducted interventions in natural settings, including forests, natural parks, urban parks, urban green areas, cultivated fields, or gardens, involving activities like walking, observing, engaging in relaxation exercises, or simply resting and breathing fresh air for a specified duration. All studies were considered to be eligible irrespective of the type of comparison or control group used.

Furthermore, the review included articles that assessed a wide range of outcomes, covering both physiological and neuropsychological measures. Within the physiological domain, we considered measurements of cortisol levels and blood pressure. In the neuropsychological category, we focused on measurements obtained using scales to assess stress, depression, and anxiety levels. Trials involving participants with pre-existing medical conditions and clinical studies utilizing virtual photographs or simulations as interventions were excluded from consideration.

### 2.2. Information Resources and Search Strategy

A structured literature search was conducted using PubMed, Scopus, and Lilacs between August and September 2021, and it included all articles published up to the search date in English and Spanish. A search for the following terms was performed: Forests, Relaxation Therapy, Nature, Humans, Walking, Psychological Stress, Physiological Stress, Anxiety, Cortisol, and Blood Pressure. The search algorithms for each database are described in Figure S1.

### 2.3. Study Selection and Data Collection Process

The search results from each database were uploaded to the Rayyan® web application. The researchers utilized the Rayyan® to screen articles. Each title and abstract was reviewed independently by at least two researchers. In cases of disagreement, a third evaluator resolved discrepancies among the researchers. Subsequently, three researchers read the full text to extract pertinent information for the study.

Data items were extracted into an Excel® spreadsheet. This information included the general characteristics of the study population, such as gender, age, and the presence of any pre-existing medical conditions. Additionally, we recorded details regarding the intervention and specific location of the natural area used for research purposes.

Regarding psychological instruments, we included studies that assessed any neuropsychological scales, including the Profile of Mood States questionnaire (POMS), Positive and Negative Affect Schedule (PANAS), Restorative Outcome Scale (ROS), Stress Response

Inventory (SRI-MF), and Perceived Stress Scale-10 items (PSS10). These scales are widely recognized for measuring psychological disorders.

### 2.4. Risk of Bias

The assessment of risk of bias adhered to the guidelines outlined in the Cochrane Manual for International Systematic Reviews [31]. Three researchers conducted the evaluation through discussion and consensus. The risk of bias parameters encompassed the randomization method employed (indicating selection bias), allocation concealment (reflecting selection bias), blinding of outcome assessment (indicating detection bias), completeness of outcome data (pertaining to attrition bias), and selective reporting (associated with reporting bias). In all instances, a positive response (+) signified a low risk of bias, while a negative response (−) signified a high risk of bias (Figure S2).

### 2.5. Strategy for Data Synthesis and Quality of Evidence Assessment

The analysis of both dichotomous and continuous outcomes was performed using the Review Manager® software (RevMan, version 5.3., Copenhagen: The Nordic Cochrane Centre, The Cochrane Collaboration, 2014) [32]. All pooled analyses were conducted utilizing the random-effects model. In instances where discrepancies arose, other researchers were consulted to resolve differences and determine the most appropriate course of action.

To identify heterogeneity, we employed the $I^2$, with percentage values of 25, 50, and 75 denoting low, moderate, and high heterogeneity, respectively [33]. For the sensitivity analysis, we followed the methodology outlined by Pértega et al. [34]. This involved systematically excluding individual studies to assess their impact on the overall findings. Consistency in the results would indicate robustness. Furthermore, we complemented this analysis with funnel plots.

## 3. Results

### 3.1. Search Results

In total, eight randomized controlled clinical trials were included (Tables 1 and S1 and Figure 1). For the physiological parameters, it was possible to identify two studies where cortisol levels were measured (Jia et al. [35] and Razani et al. [36]), one study that focused on evaluating blood pressure (Bang et al. [37]).

Regarding the use of psychological instruments, five articles were included: four studies that performed a stress assessment (Bielinis et al. [38]; Janeczko et al. [39]; Kim et al. [40,41]), five studies that focused on evaluating depression levels (Bang et al. [37]; Bielinis et al. [38]; Janeczko et al. [39]; and Kim et al. [40,41]), and five studies that focused on anxiety levels (Bielinis et al., 2018 [38]; Choe et al. [42]; Janeczko et al. [39]; and Kim et al. [40,41]). Table S1 shows the definition and main characteristics that the studies included in this review used in each psychological outcome.

The number of participants varied across the studies, ranging from 18 to 99 (media = 64; IQ = 37) (Table 1). The duration of the intervention varied across the studies, ranging from 15 min to two hours. According to the Kappa statistic obtained, there was a substantial agreement for selection by title and abstract (0.62 and 0.67, respectively).

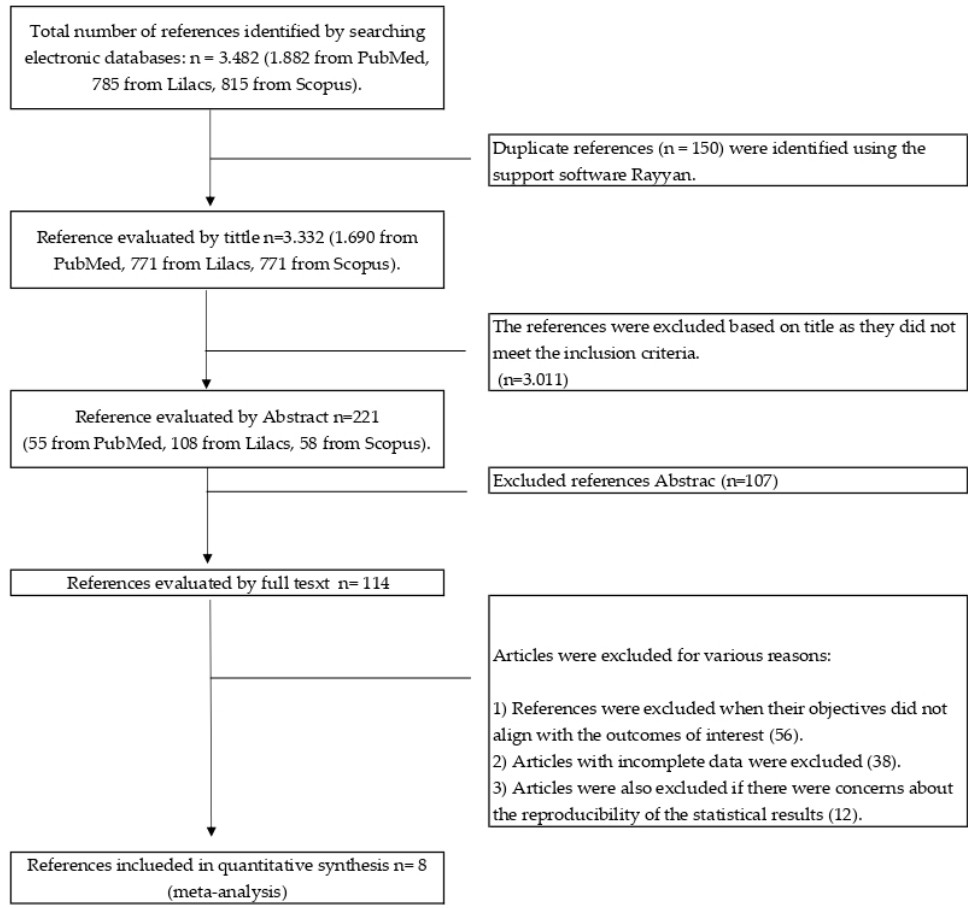

**Figure 1.** PRISMA diagram of articles selected for review.

*3.2. Risk of Bias*

Notably, none of the studies fulfilled the criteria specified in the Cochrane Manual for Systematic Reviews concerning the selection of study designs (as depicted in Figures 2 and S1–S3).

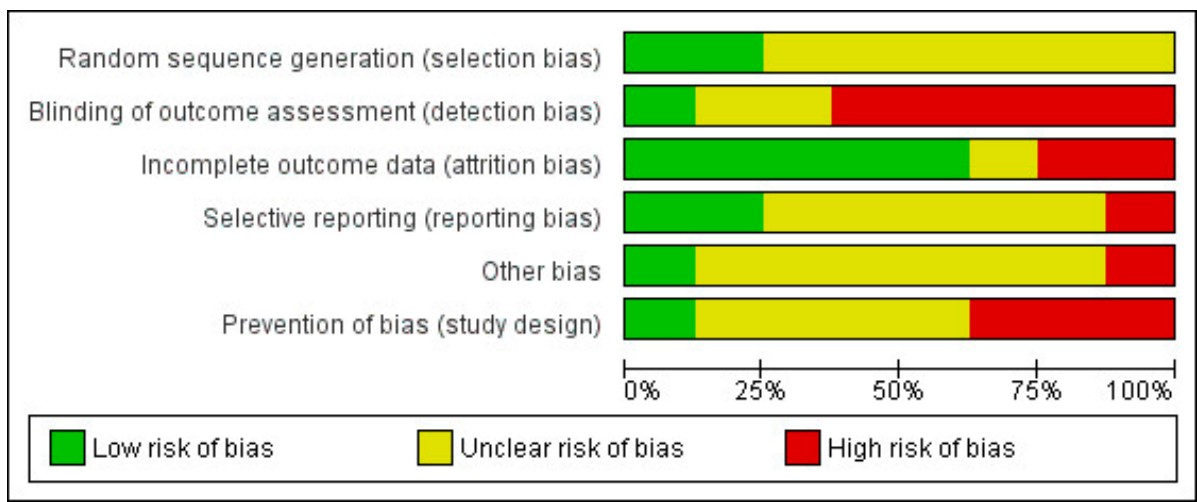

**Figure 2.** Graph of methodological quality: reviewers' evaluations on each item of methodological quality are presented as percentages for all included studies.

**Table 1.** Data from the included studies.

| Reference (Country) | N | Age Mean ± SD | Control Group | Intervention Group | Outcomes (Unit) | Duration of the Intervention | Full Description of the Intervention |
|---|---|---|---|---|---|---|---|
| Bang et al. [37] (South Korea) | 99 | 24.3 ± 4.19 | No intervention | Nature therapy | Physiological: | 6 weeks | The campus forest-walking program was conducted once per week during lunch. The university campus has many different trees, and there are nearby forest roads and trails. |
| | | | M = 21 | M = 26 | Blood pressure (mm Hg) | | |
| | | | F = 27 | F = 25 | | | |
| Bielinis et al. [38] (Poland) | 62 | 21.5 ± 0.18 | City intervention | Nature therapy | Psychological: POMS, PANAS and ROS. | 1 day | The field experiment was winter season. Two locations were selected: the urban and the forest environment (deciduous, broad-leaved urban forest situated near the city centre). |
| | | | M = 18 | M = 18 | | | |
| | | | F = 13 | F = 13 | | | |
| Jia et al. [35] (China) | 18 | 61–79 | City intervention | Nature therapy | Physiological: Cortisol (ng/mL) | 7 days | The study was performed at two different sites (nature and city). On the day before the study, blood samples were taken from the participants in the morning before breakfast. |
| | | | M/F = 8 | M/F = 10 | | | |
| Choe et al. [42] (United Kingdom) | 66 | 16–62 | Indoor environment | Natural environment | Psychological: PANAS | 6 weeks | The study consisted of an experiment of three different environments: natural outdoor, built outdoor, and indoor environments. The intervention was in groups of between 6 and 10 participants. Each weekly session lasted one hour and included mindfulness meditation/exercises and group discussion. |
| | | | M/F = 33 | M/F = 33 | | | |
| Janeczko et al. [39] (Poland) | 75 | 19–24 | Urban area | Nature therapy | Psychological: POMS, PANAS, and ROS. | 1 day | The outdoor experiment was conducted in four different settings: (1) an urban environment with a noticeably higher level of noise, (2) the scenery of urban housing, (3) the Sobieski Forest, and (4) a coniferous forest. |
| | | | M/F = 45 | M/F = 30 | Physiological: Blood pressure (mmHg). | | |
| Kim et al. [40] (South Korea) | 38 | 22 | No intervention M/F = 19 | Nature therapy | Psychological: POMS and SRI-MF | 2 months | An eight-session forest therapy program was performed once per week, and each session lasted for 1.5 h. Participants were involved in many activities, such as forest dance, forest meditation, forest exercise, walking, and others. The main purpose of the program was to reduce stress and improve the self-esteem of the participants. |
| | | | | M/F = 19 | | | |
| Kim et al. [41] (Korea) | 38 | 22.1 ± 1.6 | No intervention | Nature therapy | Psychological: POMS and SRI-MF. | 8 weeks | The participants were instructed to perform individualized, voluntary forest activities for a one hour-long session per week. The activities included stretching, breathing, walking, meditation, and exercise. |
| | | | M/F = 19 | M/F = 19 | | | |
| Razani et al. [36] (United States) | 75 | >18 | Independent park prescription | Supported park prescription | Psychological: PSS10 | 3 weeks | The outdoor experiment was conducted in three different sceneries: a bayfront park with a beach, a lake with woodlands, and a redwood forest. Outings concluded with quiet reflection and an opportunity to share experiences. |

POMS: Profile of Mood States questionnaire; PANAS: Positive and Negative Affect Schedule; ROS: Restorative Outcome Scale; SRI-MF: Stress Response Inventory; PSS10: Perceived Stress Scale—10 items; M: male; F: female; and SD: standard deviation.

### 3.3. Physiological Parameters

Two studies identified cortisol as a physiological biomarker of stress parameters. However, the data obtained between the studies could not be analysed statistically because the samples were collected from different matrices (i.e., saliva vs. blood). Razani et al. [36] conducted a study to evaluate the cortisol concentrations in saliva among a sample of 75 individuals, while Jia et al. [35] conducted a separate study in which they assessed the cortisol concentrations in the blood of 18 individuals. Nevertheless, in both studies, they concluded that there was a significant difference in the cortisol levels between the pre-test and post-tests in the intervention group (Table S2).

In one study conducted by Bang et al. [37], changes in blood pressure were assessed between an intervention group exposed to nature and a control group with no intervention. Notably, there were no significant differences ($p > 0.05$) in the blood pressure levels between the intervention group and the control group in the pre-test data for both systolic and diastolic pressure. However, a significant difference was observed in the post-test values for systolic pressure ($p = 0.001$). None of the articles included an evaluation of blood pressure using an ambulatory blood pressure monitor.

### 3.4. Psychological Parameters

3.4.1. Scales Used to Evaluate Stress

Four studies aimed at assessing stress as an outcome measure were identified [38–41]. The characteristics of the included trials involving 213 participants are summarized in Table 1. Kim et al. included 38 participants in their publications from 2020 and 2021, respectively, while Bielinis et al. included 62, and Janeczko et al. included 75 individuals.

Among those studies, two different scales were used to measure the stress response levels (Figure 3). Kim et al. (2020 [40] and 2021 [41]) used the Stress Response Inventory (SRI-MF). The SRI-MF results indicated a significant reduction in stress levels for a nature immersion therapy intervention group compared to a control group (MD = −7.87 [95% CI: −9.44, −6.30]; $I^2 = 0\%$) (Figure 3).

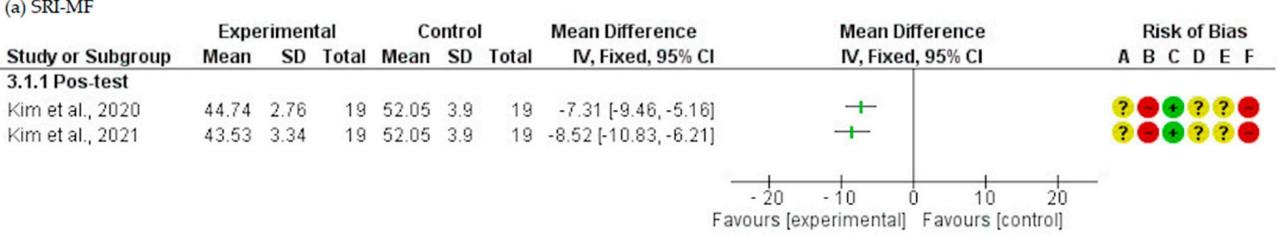

**Figure 3.** Evaluation of stress using SRI-MF and ROS scales [27,38–41].

On the other hand, Bielinis et al. [38] and Janeczko et al. [39] used the Restorative Outcome Scale (ROS). The ROS showed no significant differences between the intervention and control ($I^2$ = 96%) (Figure 3).

### 3.4.2. Scales Used to Evaluate Anxiety and Depression

Three studies were selected for the evaluation of anxiety and depression levels according to the PANAS scale, and the characteristics of the included trials involving 203 participants are summarized in Table 1. Choe et al. included 66 participants, while Bielinis et al. included 62 and Janeczko et al. included 75 individuals. However, only two studies were used in the quantitative analysis (Figure 4), because the study conducted by Choe et al. [42] used different criteria to obtain the results of this scale, and therefore, the data obtained were not comparable with the other studies (Table S1).

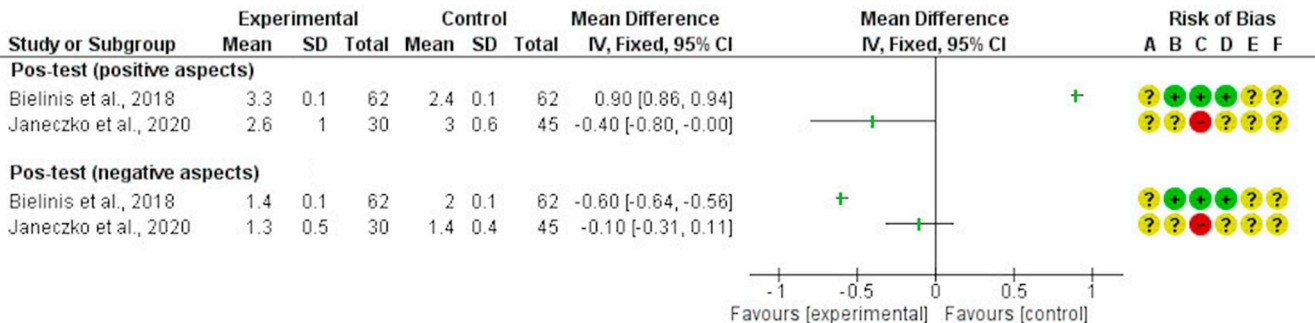

Risk of bias legend
(A) Random sequence generation (selection bias)
(B) Blinding of outcome assessment (detection bias)
(C) Incomplete outcome data (attrition bias)
(D) Selective reporting (reporting bias)
(E) Other bias
(F) Prevention of bias (study design)

**Figure 4.** Evaluation of stress anxiety and depression using PANAS scales [11,38–41].

In the quantitative analysis, the mental health and well-being outcomes exhibited higher values when assessed in natural outdoor environments compared to indoor or built environments (Figure 4). Positive aspects showed a mean value of 33.70 (95% CI: 30.84; 36.55, $p$ = 0.049), while negative aspects had a mean value of 20.12 (95% CI: 17.69; 22.55, $p$ < 0.001).

### 3.4.3. Scales Used to Evaluate Other Psychological Dimensions

Four eligible studies that used the POMS scale were identified grouping 213 participants, and their results are shown in Figure 5. Kim et al. included 38 participants in their publications from 2020 and 2021, respectively, while Bielinis et al. included 62 and Janeczko et al. included 75 individuals. It was evident that there were variations in the criteria used to evaluate the scales between the studies, which represented a limitation for a comparison between the four studies [38–41].

A subgroup analysis was performed as follows: the first subgroup contained the Bielinis et al. [38] and Janeczko et al. [39] studies, while the second subgroup contained the Kim et al. [35,36] studies. No significant difference was observed among the studies using the POMS scale ($I^2$ = 96%; Figure 5) (Kim et al. [40,41]). The intervention groups showed a significant benefit compared to the control groups ($I^2$ = 0%; Figure 5).

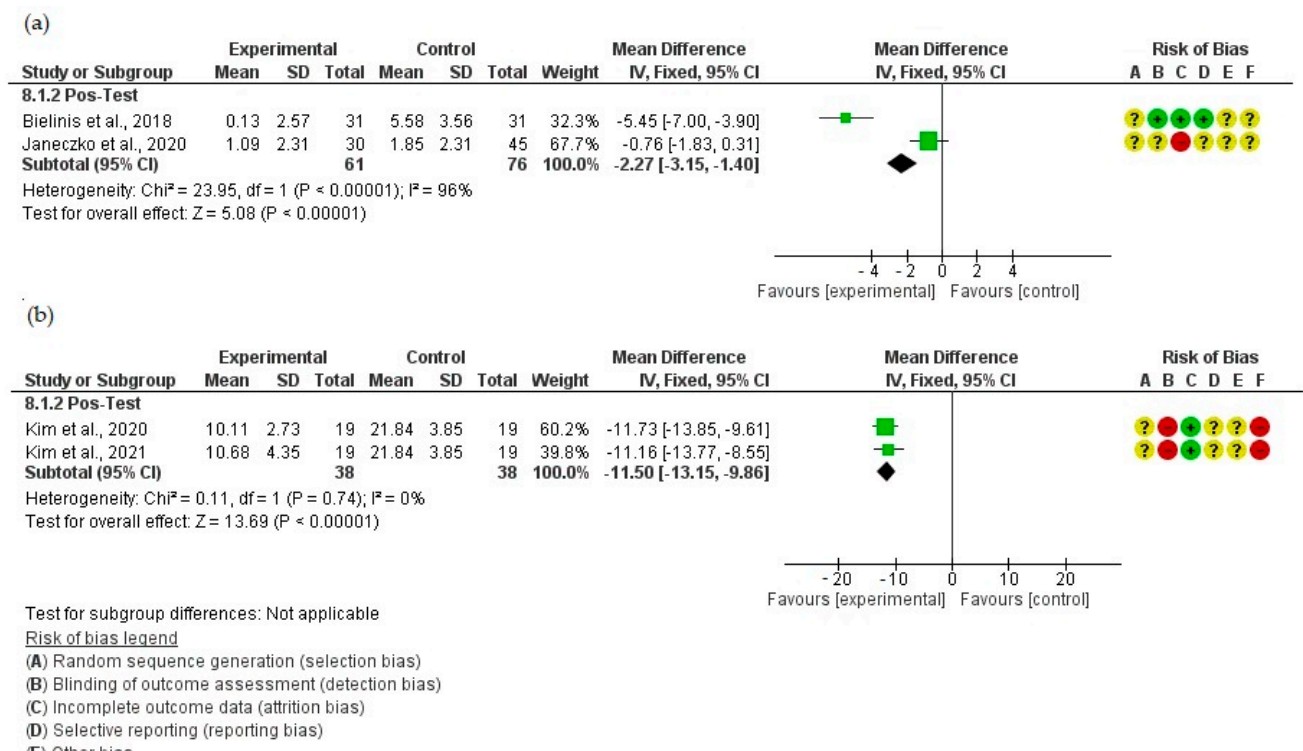

**Figure 5.** Evaluation of effect of nature intervention using POMS [24,38–41]. (**a**) The initial subgroup consisted of the studies by Bielinis and Janeczko; (**b**) the second subgroup encompassed the studies by Kim.

## 4. Discussion

Our systematic review compiles evidence from previous systematic reviews that have explored the health effects of forest-based interventions. We identified and synthesized a total of eight publications, with a predominant focus on Asian countries. However, the outcomes of this review are inconclusive, failing to provide definitive insights into the impact of these therapies on stress, anxiety, or depression levels. This lack of conclusive evidence may be attributed to a range of factors, such as variations in the types of interventions, discrepancies in sample sizes, methodological diversity in outcome assessment via different scales, and uncertainties in participant allocation, among other considerations.

In the selected studies, we observed diverse methodologies employed in the development of forest-based interventions. For instance, some studies implemented multiple sessions over several weeks, while others focused on shorter, single interventions. Additionally, the lack of clarity in reporting intervention details, such as duration and supervision, presents challenges when interpreting and comparing study results. For example, Kim et al. [40] implemented eight sessions (one per week for six weeks) with a duration from 1 to 2 h, and Kim et al. [41] showed the development of a routine (exercises, meditation, walking, and breathing) that was executed in one of the eight sessions. Bielinis et al. [38] implemented a 15 min intervention that consisted of walking in a nature area (intervention group) and walking for 15 min in an urban area (control group). That walk was only implemented once, yet Janeczko et al. [39] developed an intervention that included four different scenarios (apartments, green spaces in the suburbs, coniferous forest, and deciduous forest), for which we selected the apartment setting (control group) and the coniferous forest setting (intervention group) for the analysis. However, that study did not specify the duration of the intervention. This diversity in methodology and reporting adds complexity to drawing definitive conclusions regarding the effectiveness of forest therapies.

When discussing cortisol levels, it is essential to highlight the indiscriminate use of this biomarker without considering its circadian cycle. It is important to note that, upon

awakening, cortisol levels are expected to rise by 50–60%. Subsequently, there is a rapid decline in the ensuing hours, followed by a gradual decrease as night falls [43]. A flat cortisol rhythm, characterized by the absence of an early morning rise in cortisol levels, has been associated with depression.

In this review, the study conducted by Jia et al. [35] reported a decrease in cortisol concentration within the nature group compared to the initial measurement, with no notable differences observed in the control group. However, it is important to note that this study lacks specific methodological details concerning sample collection, including the time of the sample collection and the collection technique, which complicates the analysis of its results. Additionally, it is crucial to consider that some articles used cortisol saliva levels while others utilized cortisol blood levels. These two biological matrices are not directly comparable due to the influence of cortisol binding to proteins on serum cortisol levels.

Furthermore, for an accurate analysis, cortisol saliva levels should ideally include the first sample being taken upon awakening, along with at least a second sample collected between 30 and 45 min after awakening. None of the studies included in this review reported conducting this type of sample collection and analysis, which could indicate a potential misunderstanding of the cortisol cycle. Bang et al. [37] did not report significant differences between the control and experimental groups in the post-test analysis of blood pressure.

Regarding mental health, several reviews showed positive effects on stress, depression, and anxiety in healthy people after nature-based interventions [44–46]. In our study, the scores for the ROS scale for the Bielinis et al. [38] and Janeczko et al. [39] studies showed a favourable effect on the nature-based interventions group when compared to pre-intervention. Some review studies have shown that scores on the ROS scale increase as a result of recreation and relaxation in nature areas [45,47]. In those two studies, an increase in ROS scores was observed after walking and watching in nature areas when compared to pre-intervention scores. However, Bielinis et al. [38] noticed a decrease in ROS scores after walking and watching in urban areas when compared to pre-intervention scores.

The findings concerning the PANAS scale yielded mixed results. In one study, the positive effect was significantly higher in the nature environment, which contributed to the mental health of the participants [38]. However, Chloe et al. [42] did not report significant differences between environments for negative or positive effect. Similarly, a the second study reported that no significant differences were identified for positive effect, but there were notably decreased results observed in the post-tests for negative effect [39].

The evidence from studies employing the POMS scale is diverse. Bielinis et al. [38] administered the POMS scale 15 min before and 15 min after therapy. Janeczko et al. [39] presented their results based on three stratifications: by time, by site, and site×time. Also, the methodology used by Kim et al. [40,41] to apply the scale was not clear.

Nevertheless, in spite of the studies reporting a statistically significant reduction in the levels of anxiety and depression as measured by the POMS scale, the pooled result was not significant ($I^2 > 75$).

This review reveals certain limitations in the existing studies, manifesting a lack of consistency in their findings, possibly attributed to inadequacies in study design. For instance, among the three studies reporting PANAS results, none detailed the specific program or methodology employed for population randomization in the two study groups. Furthermore, one of these three studies failed to specify the duration of the intervention.

Additionally, a deficiency in controlling biases was observed in studies focused on nature-bathing interventions, which directly impacts the obtained results and contributes to an increased heterogeneity among the studies. In light of these observations, it becomes evident that there is a pressing need for the generation of research with stringent study designs that can yield reliable data.

Another limitation of the study is that, by exclusively focusing on a healthy population, it was not possible to directly assess the impact of nature interventions on individuals who already experience stress, anxiety, or depression. While it was suggested that these therapies might have even more pronounced benefits for individuals affected by these

conditions, this review does not provide direct evidence in that regard due to its focus on individuals without preexisting pathologies.

Kotera et al. [48] conducted a systematic review and meta-analysis of 20 studies examining the effects of Shinrin-Yoku on mental health. They found that nature immersion has a more pronounced impact on anxiety than on depression and anger. Factors such as gender and the proportion of Japanese or Asian participants were associated with greater anxiety reduction effects. While some studies showed promising results, the overall quality and potential publication bias raised concerns. The findings suggest that Shinrin-Yoku may be effective in reducing short-term mental health symptoms, particularly anxiety [48]. We concur with the authors in emphasizing the need for more rigorous research and comprehensive follow-up assessments.

Finally, the search primarily focused on academic databases, and other potential sources of information, such as clinical trial registries and previous systematic reviews, were not included. By not extending the search to these additional resources, there is a possibility that relevant evidence may have been overlooked, which could have influenced the review's findings. Additionally, the reference lists of the included articles were not thoroughly explored, which may have also resulted in the omission of important studies related to the topic.

Our review underscores the necessity for more rigorous and standardized study designs within the domain of nature-based interventions. For example, factors like temperature, humidity, and noise levels during interventions, which some authors have proposed may influence health benefits [49], were not consistently quantified or reported. To ensure the production of dependable data, future studies should offer transparent and comprehensive descriptions of interventions, including details regarding the session duration, supervision, and randomization procedures. Moreover, addressing potential biases and enhancing control over confounding variables will undoubtedly elevate the overall quality of research in this field.

It is important to note that the consulted databases may not have included work conducted in countries such as China, Korea, and Japan, where nature-based therapy interventions have a tradition spanning over 30 years. Since these regions have been pioneers in the development and application of nature-based therapies, it is plausible that there may be additional studies and evidence that have not been addressed in this review due to potential limitations in accessing data from these geographic areas. Therefore, it is advisable to consider this perspective when evaluating the entirety of the evidence in future research on nature therapies and their effects on mental health. To generate reliable data, future studies should provide clear and comprehensive descriptions of interventions, including the session duration, supervision details, and randomization procedures. Additionally, addressing potential biases and improving the control of confounding factors will enhance the quality of research in this area.

## 5. Conclusions

There is only limited evidence to recommend nature bathing for reducing stress, blood pressure, anxiety and other physiological outcomes. The quality of the evidence is poor. Further studies should be carried out in different locations, as well as include a cost efficacy analysis.

Developing an official research protocol will enhance the design of future RCT studies, as variations in study designs currently hinder the ability to arrive at conclusive findings. Moreover, registering protocols will enhance transparency in reporting crucial methodological particulars. Subsequent research endeavours should prioritize addressing these limitations to further our comprehension of the health impacts of nature therapies.

**Supplementary Materials:** The following supporting information can be downloaded: https://www.mdpi.com/article/10.3390/ejihpe14030040/s1, Figure S1: Search algorithms, Figure S2: Summary of methodological quality: reviewers' judgements about each item of methodological quality for each included study, and Figure S3: Funnel plots for a visual assessment of publication bias. Description:

funnel plot: (a) Restorative Outcome Scale (ROS); (b) Stress Response Inventory; (c) Positive and Negative Affect Schedule (positive affect); (d) Positive and Negative Affect Schedule (negative affect); and (e) Profile of Mood States. Table S1. Definition of psychological outcomes. Table S2: Cortisol study results, Jia et al. [35] (blood cortisol) and Razani et al. [36] (salivary cortisol), ng/mL.

**Author Contributions:** Conceptualization, D.M.P.-C., N.V. and J.M.-R.; methodology, D.M.P.-C., A.P.-L., E.M.T., C.P., L.B., C.G., J.C.S., J.M.-R., Y.G.T.-P. and N.V.; software, D.M.P.-C., A.P.-L., E.M.T., C.P., L.B., C.G., J.C.S., J.M.-R., Y.G.T.-P. and N.V.; validation, D.M.P.-C., A.P.-L., Y.G.T.-P., N.V. and J.M.-R.; formal analysis, D.M.P.-C., A.P.-L., Y.G.T.-P., N.V. and J.M.-R.; investigation, D.M.P.-C., A.P.-L., Y.G.T.-P., N.V. and J.M.-R.; resources, J.M.-R.; data curation, J.M.-R.; writing—original draft preparation, D.M.P.-C., A.P.-L., Y.G.T.-P., N.V. and J.M.-R.; writing—review and editing, D.M.P.-C., A.P.-L., Y.G.T.-P., N.V. and J.M.-R.; visualization, D.M.P.-C., A.P.-L., Y.G.T.-P., N.V. and J.M.-R.; supervision, J.M.-R.; project administration, J.M.-R.; funding acquisition, J.M.-R. All authors have read and agreed to the published version of the manuscript.

**Funding:** This research was funded by the Ministerio de Trabajo of Colombia, Fondo de Riesgos Laborales.

**Data Availability Statement:** No new data were created or analysed in this study. Data sharing is not applicable to this article.

**Acknowledgments:** We thank researcher Diana Pinzon for her support in decision making for the selection of the articles described in this review. Likewise, we thank the entire research team of the Environmental and Occupational Health Group of the Instituto Nacional de Salud of Colombia, for the support presented during different stages of writing this article.

**Conflicts of Interest:** The authors declare no conflicts of interest.

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
