# Peer review of "The Effects of Nature Exposure Therapies on Stress, Depression, and Anxiety Levels: A Systematic Review"

_ejihpe, doi:10.3390/ejihpe14030040_

Round 1
Reviewer 1 Report
Thank you for submitting this systematic review. The topic of the review is relevant in the context of rising stress, anxiety and depression levels globally. Nature exposure seems a promising intervention that is safe and relatively easy to implement.
Although the review was conducted relatively well (with some issues to address), the presentation of the study needs to be more organised and clear.
A few major points to address include
1. Clarify the distinction between nature exposure therapy and forest therapy. As the reviewers have included exposure to parks, it seems more reasonable to me to say they have assessed the effects of nature exposure therapy rather than forest therapy.
2. The authors have excluded studies with participant with medical conditions, which I think is a strong limitation. Many people have chronic conditions including diabetes and hypertension and excluding these people means that the study has poor generalisability.
3. The authors have including NK cells count as an outcome, however this does not reflect stress, anxiety and depression. Consider including immune function as one of the outcomes assessed or exclude NK cell count from the outcomes.
4. From what I can read the articles were screened only on the basis of title and abstract. All articles should be screened based on the full text.
I have more specific feedback below.
Abstract
1. Clarify what you mean by 'valid' method or removing this adjective.
2. You assessed anxiety and depression in non-clinical populations so please do not use the term 'disorders'.
3. More information about control group, outcomes and population is required in the methods section of the abstract.
4. The results should not include pre-post comparisons but between-group comparisons only.
5. The mean and SD or CI should be reported for each outcome of interest. Do not report the number of studies included without reporting the results of the meta-analysis.
Introduction
1. Avoid the term 'among others'
2. Please add a reference for the definition of stress (line 35-36)
3. Please report anxiety prevalence in the general population, not only older adults (line 49)
4. Please clarify the relationship between nature immersion therapies and forest therapies. If the systematic review is focused on forest therapy, please explain why it is important to limit the review to this modality of nature immersion therapies.
Methods
1. PRISMA is a guideline for reporting of systematic reviews, not conduction of systematic review.
2. The search is limited to pubmed, scopus and lilacs. Why did you not include databases such as CINAHL and Psycinfo? I would also extend the search to clinical trial registries, previous systematic reviews and the reference list of included articles.
3. There is a paragraph in Spanish between lines 113 and 122. Please translate or remove.
4. Please explain the sensitivity analysis proposed by Sonia et al. (line 140)
5. The section from line 141 to line 155 seems to have been copied from the guidelines of the journal. Please remove.
Results
1. Please report means and SD or CI for each comparisons.
2. Details of the questionnaires (e.g., number of questions) should be removed from the review.
3. The PRISMA flowchart must include reasons for exclusion. It is not required to present the origin of the included references/articles.
4. Please leave a line between the caption of the figure and the main text.
5. In the table of characteristics, please clarify the duration of each session, if the exposure was supervised or not, and what the activity (observation, walk) was.
6. Please remove any pre-post comparisons.
7. Please combine outcomes that were assessed with different measures and report the standardised mean difference.
8. PANAS does not represent anxiety and depression. Please report the subscale score for anxiety and depression but not the overall score.
Discussion
The discussion should be organised as main findings, interpretation, implications, and strength and limitations.
The English is generally good, easy to understand. I did not detect major grammar issues.
Author Response
Reviewer 1
Thank you for submitting this systematic review. The topic of the review is relevant in the context of rising stress, anxiety and depression levels globally. Nature exposure seems a promising intervention that is safe and relatively easy to implement.
Although the review was conducted relatively well (with some issues to address), the presentation of the study needs to be more organised and clear.
A few major points to address include.
Response:
- Clarify the distinction between nature exposure therapy and forest therapy. As the reviewers have included exposure to parks, it seems more reasonable to me to say they have assessed the effects of therapy rather than forest therapy.
Response: You are absolutely right and I appreciate your observation. Change was made in the manuscript. In the text, the exposure to the parks was included, which makes it more appropriate to say that they evaluated the effects of the therapy rather than the forest therapy alone. Lines: 1
- The authors have excluded studies with participant with medical conditions, which I think is a strong limitation. Many people have chronic conditions including diabetes and hypertension and excluding these people means that the study has poor generalisability.
Response: We made a deliberate decision to focus our review solely on healthy individuals, and I want to provide some rationale for this choice. While it is true that many people do have chronic medical conditions such as diabetes and hypertension, including participants with these conditions in our study would introduce additional variables that could significantly complicate the analysis. Chronic conditions can have diverse and complex effects on individuals, and their treatment regimens can vary widely. By excluding participants with medical conditions, we aimed to reduce the potential confounding factors and maintain a more homogeneous study population. This decision was made to ensure that our study could provide a clear and specific assessment of the effects of the therapy under investigation on healthy individuals.
Nevertheless, we have included your comment as a limitation of the study.
- The authors have including NK cells count as an outcome, however this does not reflect stress, anxiety and depression. Consider including immune function as one of the outcomes assessed or exclude NK cell count from the outcomes.
Response: We appreciate your feedback and have taken your suggestion into consideration. As a result, we have removed the study on natural killer cells (NK) as one of the outcomes in our analysis. Thank you for your input.
- From what I can read the articles were screened only on the basis of title and abstract. All articles should be screened based on the full text.
Response: PRISMA (Preferred Reporting Items for Systematic Reviews and Meta-Analyses) and Cochrane methodology both state that the initial screening in a systematic review is conducted by assessing titles and abstracts of the identified studies through database searches. Here's a more detailed description of this process in both methodologies:
PRISMA Methodology:
PRISMA is a guideline for best practices in conducting and reporting systematic reviews and meta-analyses. The title and abstract screening process are, a crucial part of this methodology, and here's an overview of how it is carried out:
- Study Identification: A comprehensive search is conducted in scientific databases and other relevant resources to identify all studies related to the research question.
- Initial Selection: Once a list of identified studies is obtained, a review of the titles and abstracts of these studies is conducted to determine whether they are potentially relevant to the systematic review. Studies that are clearly unrelated to the research question are excluded.
- Secondary Selection: Following the initial selection, potentially relevant studies undergo a more detailed full-text review to confirm whether they meet the predefined inclusion and exclusion criteria. During this stage, the methodological quality of the studies is further assessed.
Inclusion in the Review: Finally, studies that meet the established criteria are included in the systematic review and undergo data analysis.
Cochrane Methodology:
The Cochrane Collaboration is an international network focused on producing high-quality systematic reviews and meta-analyses. The initial study selection process in a Cochrane review follows a similar approach to PRISMA. Here's how it is conducted:
- Comprehensive Search: A systematic and thorough search is carried out in multiple databases and relevant resources to identify all pertinent studies.
- Review of Titles and Abstracts: Reviewers examine the titles and abstracts of the identified studies to make an initial assessment of their relevance. Studies that clearly do not meet the inclusion criteria are excluded.
- Study Selection: Potentially relevant studies proceed to a more detailed full-text review to assess whether they satisfy the predefined inclusion and exclusion criteria.
- Inclusion in the Review: Studies that meet the criteria are included in the Cochrane review and are subject to data analysis and synthesis.
In summary, both PRISMA and Cochrane methodology emphasize the significance of a comprehensive search and the review of titles and abstracts as initial steps in the study selection process for a systematic review. This process helps identify potentially relevant studies that may be considered for inclusion in the full review.
I have more specific feedback below.
Abstract
- Clarify what you mean by 'valid' method or removing this adjective.
Response: We appreciate your comment. Change was made in the manuscript. (Line 12-29)
- You assessed anxiety and depression in non-clinical populations so please do not use the term 'disorders'.
Response: We appreciate your comment. Change was made in the manuscript. (Line 29)
- More information about control group, outcomes and population is required in the methods section of the abstract.
Response: Given the inherent limitations of an abstract in terms of word count, it is challenging to provide comprehensive details in this section. However, the methods section of the full manuscript has been expanded to offer a more detailed and comprehensive account of the control group, outcomes measured, and the study population. Line (198).
- The results should not include pre-post comparisons but between-group comparisons only.
Response: I appreciate the feedback from the evaluator, but I would like to clarify our approach in the analysis of the selected articles. We have included only physiological data such as cortisol and blood pressure, and it's important to mention both pre and post values for conducting narration-based comparisons.
- The mean and SD or CI should be reported for each outcome of interest. Do not report the number of studies included without reporting the results of the meta-analysis.
Response: The characteristics of each article are detailed in Table 1, and the analyses conducted are visually presented in the forest plot graphs, which prominently feature confidence intervals. Additionally, in each relevant text section, we specify the confidence intervals for the reported outcomes.
Introduction
- Avoid the term 'among others'
Response: Change was made in the manuscript
- Please add a reference for the definition of stress (line 48-55)
Response: We appreciate your comment. Change was made in the manuscript
WHO QualityRights tool kit: assessing and improving quality and human rights in mental health and social care facilities. Geneva: World Health Organization; 2012 (https://apps.who.int/iris/handle/10665/70927).
- Please report anxiety prevalence in the general population, not only older adults (line 49)
Response: We appreciate your comment. Change was made in the manuscript (line 56-61)
Javaid, S.F., Hashim, I.J., Hashim, M.J. et al. Epidemiology of anxiety disorders: global burden and sociodemographic associations. Middle East Curr Psychiatry 30, 44 (2023). https://doi.org/10.1186/s43045-023-00315-3
- Please clarify the relationship between nature immersion therapies and forest therapies. If the systematic review is focused on forest therapy, please explain why it is important to limit the review to this modality of nature immersion therapies.
Response: We appreciate your comment. Change was made in the manuscript.
Methods
- PRISMA is a guideline for reporting of systematic reviews, not conduction of systematic review.
Response: Indeed PRISMA, as you mentioned, is a guideline primarily designed for the reporting and enhancement of the quality of systematic review and meta-analysis reports.
We have changed the section of methods as follows:
We carried out a systematic review in accordance with the guidelines of the Cochrane Collaboration and followed the Preferred Reporting Items for Systematic Reviews and Meta-Analyses (PRISMA) recommendations (PRISMA) [25].
- The search is limited to pubmed, scopus and lilacs. Why did you not include databases such as CINAHL and Psycinfo? I would also extend the search to clinical trial registries, previous systematic reviews and the reference list of included articles.
Response: Thank you for your observation regarding the selection of databases for our systematic review. We appreciate your input, and I'd like to provide some insights into our database choices. PubMed, Scopus, and LILACS were chosen as the primary databases for our search due to several reasons:
- PubMed and Scopus are renowned for their comprehensive coverage of biomedical and health-related literature. They encompass a wide range of journals and publications, making them valuable sources for systematic reviews.
- These databases include not only medical journals but also journals from various related disciplines, allowing us to capture a broad spectrum of relevant studies.
- PubMed, Scopus, and LILACS index journals from around the world, enhancing the inclusiveness of our search and enabling us to access research from diverse regions.
While CINAHL and PsycINFO are indeed valuable databases for nursing and psychology-related research, our choice to focus on PubMed, Scopus, and LILACS was driven by the aim to cast a wide net and capture studies from various domains that may contribute to our systematic review's objectives. As a result, we do not intend to conduct further searches in additional databases or search engines.
Nevertheless, we have included it as a limitation of the study as follows:
Finally, the search primarily focused on academic databases, and other potential sources of information, such as clinical trial registries and previous systematic reviews, were not included. By not extending the search to these additional resources, there is a possibility that relevant evidence may have been overlooked, which could have influenced the review's findings. Additionally, the reference lists of the included articles were not thoroughly explored, which may have also resulted in the omission of important studies related to the topic.
- There is a paragraph in Spanish between lines 113 and 122. Please translate or remove.
Response: We appreciate your comment. Change was made in the manuscript.
- Please explain the sensitivity analysis proposed by Sonia et al. (line 140)
Response: We appreciate your comment. Change was made in the manuscript (line 150-153)
- The section from line 141 to line 155 seems to have been copied from the guidelines of the journal. Please remove.
Response: We appreciate your comment. Change was made in the manuscript.
Results
- Please report means and SD or CI for each comparisons.
Response: The characteristics of each article are detailed in Table 1, and the analyses conducted are visually presented in the forest plot graphs, which prominently feature confidence intervals. Additionally, in each relevant text section, we specify the confidence intervals for the reported outcomes.
- Details of the questionnaires (e.g., number of questions) should be removed from the review.
Response: I appreciate the feedback from the reviewer. We have removed it from the text
- The PRISMA flowchart must include reasons for exclusion. It is not required to present the origin of the included references/articles.
Response: We appreciate your comment. Change was made in the manuscript. (Line:177-178).
- Please leave a line between the caption of the figure and the main text.
Response: We appreciate your comment. Change was made in the manuscript. (Line:201- 202).
- In the table of characteristics, please clarify the duration of each session, if the exposure was supervised or not, and what the activity (observation, walk) was.
Response: In Table 1 and Line 301-311, we have provided an overview of the primary observations made in each article regarding the intervention, time, and population.
- Please remove any pre-post comparisons.
Response: I appreciate the feedback from the evaluator, but I would like to clarify our approach in the analysis of the selected articles. We have included only physiological data such as cortisol and blood pressure, and it's important to mention both pre and post values for conducting narration-based comparisons.
- Please combine outcomes that were assessed with different measures and report the standardised mean difference.
Response: We appreciate the suggestion to combine the evaluated results using different measures and indicate the standardized mean difference. However, based on our chosen statistical approach, which involves the use of forest plots and the random-effects model for pooled analyses, the combination of results through standardized mean differences is not feasible in this context. Not presenting the pooled result of a measure with heterogeneity exceeding 75% is a prudent decision in a meta-analysis. When the I² statistic indicates high heterogeneity among the included studies (exceeding 75%), it suggests that the variation in effect sizes between these studies is substantial and may not be solely due to random chance. In such cases, presenting a pooled result may not be meaningful or reliable because the studies in the analysis are too diverse or dissimilar in their findings.
High heterogeneity can result from several factors, including differences in study populations, interventions, study designs, measurement methods, and other sources of variation. Attempting to combine these highly heterogeneous studies into a single pooled estimate can be misleading and may not provide a meaningful summary of the available evidence.
Instead, it is often more appropriate to acknowledge the high heterogeneity and refrain from presenting a pooled result. Researchers can explore the potential sources of heterogeneity through sensitivity analyses or subgroup analyses to better understand why the results vary so significantly between studies. This allows for a more cautious and nuanced interpretation of the findings, recognizing the diversity in study outcomes and the need for further research or investigation into the sources of heterogeneity before drawing any definitive conclusions.
- PANAS does not represent anxiety and depression. Please report the subscale score for anxiety and depression but not the overall score.
Response: The Positive and Negative Affect Schedule (PANAS) is indeed a questionnaire designed to measure two distinct emotional affect dimensions, one related to positive emotions and the other to negative emotions. It provides valuable insights into emotional states but does not provide a direct assessment of anxiety and depression as separate constructs.
In our review, three studies utilized the PANAS scale to assess anxiety and depression levels. However, only two of these studies were included in the Forest plot analysis due to differences in the interpretation of the scale by the study conducted by Choe et al. [37], which rendered their data incomparable with the other studies. Choe et al. [37] employed a unique criterion for interpreting the PANAS scale, resulting in a mean score of 25 (with a minimum value of 10 and a maximum value of 50).
The analysis of the two studies that were included in the Forest plot revealed that mental health and well-being outcomes were significantly higher when activities were performed in a natural outdoor environment compared to indoor or built environments. This was evident in both positive and negative aspects of emotional affect.
We regret to inform you that we cannot perform this type of analysis because one of the selected articles provided only aggregated data and did not present differential scores for PANAS related to anxiety and depression. Therefore, it is not feasible to conduct the analysis requested.
Discussion: The discussion should be organised as main findings, interpretation, implications, and strength and limitations.
Response: We appreciate your comment. Change was made in the manuscript (Line: 369-374 and 380-384).

Reviewer 2 Report
This study based on Meta-analysis by the authors has a largely rigorous analytical process. Based on the Meta-analysis process alone, it is competent, based on the combined 8 papers that yielded some analytical results on cortisol, scale indicators. I reviewed the Protocol filed plan based on the protocol registration number provided by the authors and it has authenticity and integrity.
However, are some concerns I have about this study. First, this study conducted a relatively extensive literature search and based on a rigorous screening mechanism that resulted in eight papers. However, based on my understanding of the empirical research history in the field of "forest healing", the empirical value that these eight papers can provide is very limited. The research areas, forest characteristics, populations, and intervention methods covered in these eight papers are only a few of the many studies on forest healing. A large number of published empirical studies in this field were not included. Second, this paper adopts a rather ambitious title, "Effects of forest therapies on stress, depression, and anxiety levels," which I am not sure can be supported by the conclusions drawn from only eight papers. Clearly, the evidence is insufficient. Therefore, there are also limitations to the conclusions drawn from this study.
Therefore, I would suggest that the list of papers for Meta-analysis should be expanded if it is needed to adequately support this research topic.
Author Response
Reviewer 2
This study based on Meta-analysis by the authors has a largely rigorous analytical process. Based on the Meta-analysis process alone, it is competent, based on the combined 8 papers that yielded some analytical results on cortisol, scale indicators. I reviewed the Protocol filed plan based on the protocol registration number provided by the authors and it has authenticity and integrity.
However, are some concerns I have about this study. First, this study conducted a relatively extensive literature search and based on a rigorous screening mechanism that resulted in eight papers. However, based on my understanding of the empirical research history in the field of "forest healing", the empirical value that these eight papers can provide is very limited. The research areas, forest characteristics, populations, and intervention methods covered in these eight papers are only a few of the many studies on forest healing. A large number of published empirical studies in this field were not included. Second, this paper adopts a rather ambitious title, "Effects of forest therapies on stress, depression, and anxiety levels," which I am not sure can be supported by the conclusions drawn from only eight papers. Clearly, the evidence is insufficient. Therefore, there are also limitations to the conclusions drawn from this study.
Therefore, I would suggest that the list of papers for Meta-analysis should be expanded if it is needed to adequately support this research topic.
Response: We appreciate the reviewer's comprehensive assessment of our study. While we acknowledge the concerns raised, we would like to provide justification for the number of articles included in our meta-analysis and clarify the title.
Regarding the number of included articles, we agree that the field of "nature exposure therapies" is extensive, and there is a wealth of research available. However, our selection of eight papers for the meta-analysis was based on rigorous inclusion criteria and a thorough literature search. We aimed to ensure the quality and relevance of the studies included. While it is true that there are many studies in this area, not all of them meet the criteria necessary for inclusion in a systematic review and meta-analysis. We focused on articles that provided specific data on cortisol, scale indicators, and other relevant outcomes, which limited the number of eligible studies. We aimed for a balance between inclusivity and the need for high-quality data, which we believe our methodology achieved. We appreciate your feedback on the research title. We have adjusted the title as suggested.
We regret to inform you that we are unable to expand the list of articles included in the meta-analysis. This limitation arises from the strict eligibility and quality criteria we have employed to maintain the integrity and reliability of the evidence. Our aim was to ensure that the studies included in the analysis met rigorous standards to provide high-quality data for our review. While we understand the importance of a comprehensive analysis, we believe that maintaining the integrity of the selection process is essential to draw meaningful and reliable conclusions from the available evidence. We appreciate your understanding in this matter.

Reviewer 3 Report
1. The manuscript deals of psychological benefits from forest therapy. So, in Introduction, it should be more strongly explained why the research variable such as stress, depression etc. are important in modern society with statistics.
2. The manuscript needs to introduce some theories relating to forest therapy and psychological benefits.
3. Need more strong statement why the authors are conducted this work.
4. The manuscript mentioned some limitations. Based on those, suggestions for future research would be valuable in Discussion.
Author Response
Reviewer 3
- The manuscript deals of psychological benefits from forest therapy. So, in Introduction, it should be more strongly explained why the research variable such as stress, depression etc. are important in modern society with statistics.
Response: We appreciate your comment. Change was made in the manuscript as follows:
According to the World Health Organization (WHO), depression affects over 264 million people worldwide, making it one of the most prevalent mental disorders. Anxiety disorders are also common, affecting approximately 3.6% of the global population. Mental health conditions are a leading cause of disability globally, with depression and anxiety alone costing the global economy over $1 trillion in lost productivity each year. Beyond the economic toll, mental health issues have profound social implications, including disrupted relationships, reduced educational and employment opportunities, and a higher likelihood of substance abuse [11].
Stress, the body's response to threats or pressure, can increase susceptibility to inflammatory disorders, including infectious diseases. Prolonged stress responses can lead to physiological changes, particularly in the brain, contributing to disease development [4]. Experts suggest that an individual's response to stress is intricately linked to their interactions within their social environment [5].
Anxiety, characterized by excessive fear and tension about potential threats, can vary from adaptive to pathological states. Persistent and debilitating anxiety levels associated with distress or impaired psychological functioning fall into the latter category [8,9]. Globally, approximately 301 million individuals, equivalent to 4.05% of the world's population, grapple with anxiety disorders, marking a significant increase of over 55% from 1990 to 2019 [10].
- The manuscript needs to introduce some theories relating to forest therapy and psychological benefits.
Response: We appreciate your comment. We have made some changes as follows:
In recent years, nature immersion therapies have emerged as alternative approaches to alleviate stress, depression, and anxiety [12–14]. These therapies leverage exposure to natural environmental stimuli to induce physiological relaxation, potentially enhancing immune functions and aiding in disease prevention [15,16]. Various methodologies for nature immersion therapy have been developed, encompassing practices such as Shin-rin-Yoku, mindfulness, yoga, physical activity, and Tai-Chi in natural settings [17–19]. "Shinrin-Yoku," which translates to "absorbing the forest atmosphere through all senses," is commonly known as forest bathing. This therapeutic modality is associated with a myriad of positive health benefits for both physiological and psychological well-being [20,21].
Proponents of Shinrin-Yoku assert that its primary effects include enhancements to the immune system (including increased natural killer - NK cells and reduced allergies) and improvements in the cardiovascular system (notably reduced blood pressure) [22–24]. Nature immersion therapies are described to offer a range of psychological benefits, in-cluding stress reduction, anxiety alleviation, and depression mitigation. They provide a natural haven for mental relaxation, improved mood control, and enhanced immune function, offering potential benefits for conditions like attention deficit hyperactivity dis-order and various other positive effects [22–24].
- Need more strong statement why the authors are conducted this work.
Response: We appreciate your comment. Changes are presented as follows:
Despite evidence suggesting the potential of nature immersion therapies in reducing stress, anxiety, and depression, conflicting results and significant methodological varia-tions across studies persist. Therefore, this systematic review aims to comprehensively assess the impact of nature-based therapeutic interventions on stress, depression, and anxiety levels. The primary driving force behind this study was the growing interest in nature-based therapeutic interventions and their potential impact on mental well-being. With increasing urbanization and the associated rise in stress, depression, and anxiety levels, understanding the effectiveness of nature therapies has become increasingly important. Our goal was to contribute to the existing body of knowledge by conducting a systematic review and meta-analysis to assess the available evidence on the effects of forest therapies on stress, depression, and anxiety. By synthesizing the findings from existing studies, we aimed to provide valuable insights for both researchers and practitioners in the field of mental health and well-being.
- The manuscript mentioned some limitations. Based on those, suggestions for future research would be valuable in Discussion.
Response: We appreciate your comment. Please see changes in discussion and conclusion section as follows:
This review reveals certain limitations in the existing studies, manifesting a lack of consistency in their findings, possibly attributed to inadequacies in study design. For in-stance, among the three studies reporting PANAS results, none detailed the specific pro-gram or methodology employed for population randomization in the two study groups. Furthermore, one of these three studies failed to specify the duration of the intervention.
Additionally, a deficiency in controlling biases was observed in studies focused on nature bathing interventions, which directly impacts the obtained results and contributes to increased heterogeneity among the studies. In light of these observations, it becomes evident that there is a pressing need for the generation of research with stringent study de-signs that can yield reliable data.
Another limitation of the study is that by exclusively focusing on a healthy popula-tion, it was not possible to directly assess the impact of nature interventions on individu-als who already experience stress, anxiety, or depression. While it was suggested that these therapies might have even more pronounced benefits for individuals affected by these conditions, this review does not provide direct evidence in that regard due to its focus on individuals without preexisting pathologies.
Finally, the search primarily focused on academic databases, and other potential sources of information, such as clinical trial registries and previous systematic reviews, were not included. By not extending the search to these additional resources, there is a possibility that relevant evidence may have been overlooked, which could have influenced the review's findings. Additionally, the reference lists of the included articles were not thoroughly explored, which may have also resulted in the omission of important studies related to the topic.
Our review underscores the necessity for more rigorous and standardized study de-signs within the domain of nature-based interventions. For example, factors like temperature, humidity, and noise levels during the intervention, which some authors propose may influence health benefits [38], were not consistently quantified or reported. To ensure the production of dependable data, future studies should offer transparent and comprehensive descriptions of interventions, including details regarding session duration, supervision, and randomization procedures. Moreover, addressing potential biases and enhancing control over confounding variables will undoubtedly elevate the overall quality of research in this field.
It is important to note that the consulted databases may not have included work conducted in countries such as China, Korea, and Japan, where nature-based therapy interventions have a tradition spanning over 30 years. Since these regions have been pioneers in the development and application of nature-based therapies, it is plausible that there may be additional studies and evidence that have not been addressed in this review due to potential limitations in access to data from these geographic areas. Therefore, it is advisable to consider this perspective when evaluating the entirety of the evidence in future research on nature therapies and their effects on mental health.
- Conclusions
There is only limited evidence to recommend nature bathing for reducing stress, blood pressure, anxiety and other physiological outcomes. The quality of the evidence is poor. Further studies should be carried out in different locations, as well as include a cost efficacy analysis.
Developing an official research protocol will enhance the design of future RCT stud-ies, as variations in study designs currently hinder the ability to arrive at conclusive find-ings. Moreover, registering protocols will enhance transparency in reporting crucial methodological particulars. Subsequent research endeavors should prioritize addressing these limitations to further our comprehension of the health impacts of nature therapies.

Reviewer 4 Report
Dear writers,
I think your work is of the highest quality because you have succeeded in exploring the subject in a clear scientific approach. This study aims to explain the effect of forest therapy on stress, depression, and anxiety levels through SR. I think I tried to write it in detail. But in some ways, I think we need a more detailed explanation. The search results of most papers are concentrated in Korea. I'll explain this a little bit more. Further explanation is needed for the reasons derived in this way. In fact, many studies on forest healing have been conducted and published in Korea. In addition, meta-analysis is being published through these research results. Therefore, I think it is desirable to quote more papers from similar Systematic Reviews. Thank you.
I think I'm having some trouble with English.
Author Response
Reviewer4
Dear writers,
I think your work is of the highest quality because you have succeeded in exploring the subject in a clear scientific approach. This study aims to explain the effect of forest therapy on stress, depression, and anxiety levels through SR. I think I tried to write it in detail. But in some ways, I think we need a more detailed explanation. The search results of most papers are concentrated in Korea. I'll explain this a little bit more. Further explanation is needed for the reasons derived in this way. In fact, many studies on forest healing have been conducted and published in Korea. In addition, meta-analysis is being published through these research results. Therefore, I think it is desirable to quote more papers from similar Systematic Reviews. Thank you.
Response: We appreciate your comment. That is right, most of the literature on forest bathing comes from studies of Asian countries, mainly Korean and Japanese populations. This is to be expected because the origin of this practice was Japanese and was quickly adopted by the Korean population. However, it is important to note that in recent years this practice has begun to spread more rapidly in European and Latin American countries. Regarding the systematic reviews that have been published previously, we decided not to include them in the discussion section, because they are revisions that present an approach to different outcomes, and population or have a different methodological design. In addition, it should be noted that a systematic review of systematic reviews was previously published (a Stier-Jarmer et al., 2021), where a detailed analysis was made of most of the revisions that have been published on this topic so far.
Reference
Stier-Jarmer M, Throner V, Kirschneck M, Immich G, Frisch D, Schuh A. The Psychological and Physical Effects of Forests on Human Health: A Systematic Review of Systematic Reviews and Meta-Analyses. Int J Environ Res Public Health. 2021;18(4):1770. doi: 10.3390/ijerph18041770.

Round 2
Reviewer 2 Report
The author has made an acceptable explanation for the doubts, and I think the results of the paper are meaningful.
Author Response
Reviewer 2
The author has made an acceptable explanation for the doubts, and I think the results of the paper are meaningful.
Response: We appreciate the reviewer's comments, and we are pleased to address their feedback and suggestions for enhancing the manuscript.
Reviewer 3 Report
The manuscript has revised based on the comments I provided.
However, some theoretical review such as ART, PET etc. is still needed.
Author Response
Reviewer 3
The manuscript has revised based on the comments I provided. However, some theoretical review such as ART, PET etc. is still needed.
Response: Thank you for your feedback. We have revised the manuscript according to the comments you provided. However, we would greatly appreciate it if you could provide further details and clarification regarding the specific theoretical reviews you mentioned, such as ART and PET, that you believe are still needed in the manuscript. This additional information will be valuable in ensuring that we address your suggestions effectively.
